# What the Upper Atmospheres of Giant Planets Reveal

**James O'Donoghue** [1],*[ID] **and Tom Stallard** [2][ID]

1   Department of Solar System Science, JAXA Institute of Space and Astronautical Science, Sagamihara 252-5210, Japan

2   Department of Mathematics, Physics and Electrical Engineering, Northumbria University, Newcastle-upon-Tyne NE1 8ST, UK

*   Correspondence: jameso@ac.jaxa.jp

**Abstract:** The upper atmospheres of the Giant Planets, Jupiter, Saturn, Uranus and Neptune are transition regions between meteorological layers and outer space. As a result of their exceptionally rarefied nature, they are highly sensitive and therefore revealing probes of the forcing exerted both from above and below. This review provides an overview of these upper atmospheres and the major processes that take place within them, including their powerful auroras, the giant planet 'energy crisis' and the decay of Saturn's rings into the planet. We discuss the many remote-sensing tools that have been used to understand them, for example, large ground-based observatories such as the Keck telescope, space-based observatories such as the Hubble Space Telescope and orbiters such as the Cassini spacecraft. Looking into the future, we discuss the possibilities afforded by the latest and next generation of observatories and space missions, such as the James Webb Space Telescope.

**Keywords:** space; planets; atmospheres; ionospheres; aurorae; rings; solar wind

## 1. Background and Motivation

The habitability of planets, as far as we know, depends chiefly on the conditions of their atmospheres. Atmospheric escape rates, which positively correlate with temperature, determine the stability of atmospheres in the past, present and future. As this escape happens in the upper atmosphere, an understanding of the thermal conditions and major processes in the region is crucial for understanding the long-term evolution of atmospheres in general, both in the solar system and beyond.

The solar system is host to the four giant planets Jupiter, Saturn, Uranus and Neptune. Jupiter and Saturn are commonly referred to as gas giants while Uranus and Neptune are known as ice giants; this terminology is understandably biased towards the chemical and thermal conditions found of the visible atmosphere, though these worlds in their bulk composition are composed of fluids in a supercritical state. The upper atmospheres of giant planets typically begin hundreds of kilometers above the 'visible surface' seen in most images and end near the vacuum of space. More specifically, upper atmospheres begin where turbulent mixing no longer dominates and each atom and molecule is able to separate according to its weight under gravity, leading to lighter constituents such as molecular and atomic hydrogen dominating the composition of upper atmospheres. Therefore, despite each giant planet having its own unique bulk composition, all of their upper atmospheres share a broadly similar chemical make-up.

Signs of activity in giant planet upper atmospheres were found as early as the 1950s, with the discovery of radio emissions emanating from Jupiter [1]. These radio emissions are cyclotron emissions, generated by electrons, which are deflected by magnetic fields, thus they were a clear sign of magnetospheric activity. They have been studied in increasing detail ever since by examining emissions across the electromagnetic spectrum from radio waves to X-rays using ground-based telescopes, sounding rocket experiments, Earth-orbiting telescopes and by spacecraft either flying-by each planet or orbiting them [2–4].

Methods for remotely sensing the upper atmosphere are not limited to examining emissions; however, their properties can be ascertained by observing how they attenuate electromagnetic waves from natural (the sun and stars) or artificial sources (spacecraft radio emissions) as they pass behind the planetary atmosphere from an observer's perspective. This review focuses mainly on observations made in the infrared and ultraviolet, as emissions at those wavelengths constitute over 95% of emitted power from upper atmospheres [5].

Figure 1 shows Jupiter, Saturn and Uranus from a variety of remote-sensing tools based on/near Earth or near each planet at infrared and ultraviolet wavelengths. The aurorae of Jupiter and Saturn at the poles are over 10 times more emissive than the background non-auroral atmosphere in IR and UV, so most of what we have learned about upper atmospheres has been obtained through studies of these highly active regions [2,6,7]. The aurorae of Uranus are approximately 10 times brighter than the disk emissions of the planet in UV, but only amount to 25% of the emission from the planet's upper atmosphere in infrared [8–10]. Notably absent from Figure 1 is Neptune. Weak upper-atmospheric UV emissions were recorded by the Voyager 2 spacecraft during a flyby of the planet in 1989 [9], but no images comparable to that of Uranus were reconstructed from this data, likely due to a combination of weak emissions and low spatial resolution.

The intensity of light received at Earth from each giant planet falls as $1/r^2$, as does the solid angle presented in our sky. By taking account of the different physical sizes of each planet but assuming equal emissions from each, the amount of light observed from Saturn, Uranus and Neptune is 18%, 0.7% and 0.3% as intense relative to Jupiter. However, as mentioned above, the aurorae of Jupiter-to-Neptune diminish by three orders of magnitude, so our knowledge of each planet's upper atmosphere is similarly scaled. The Voyager 1 and 2 spacecraft brought an identical laboratory to each giant planet, somewhat leveling the playing field in the 1980s, but as no subsequent missions have visited the ice giants, we are limited to Earth-based observations and the diminishing returns from the inverse-square law. These facts underscore the need for new planetary space probes, larger telescopes and more sensitive instrumentation if we are to understand the ice giants to the same degree as the inner giants.

Most of this review focuses on what we have learned at Jupiter and Saturn; the closest, brightest and largest of the four giant planets, which have been visited by multiple spacecraft. Owing to their proximity and physical size, we have been able to observe them in great detail, allowing us to study global phenomena such as Jupiter and Saturn's aurora-transmitting global-scale waves of heat toward the equator, or unique interactions such an in-fall of material from the rings of Saturn known as 'ring rain'. In our review of Uranus and Neptune, and using our knowledge of the physical processes at Jupiter and Saturn as a reference, we discuss what we might learn with new and future observatories and space probes. Beyond the solar system, over 5000 exoplanets have been confirmed orbiting other stars, with thousands more candidates awaiting confirmation (at the time of writing), so we will also discuss the possibility of observing exoplanetary upper atmospheres and highlight what such observations could reveal.

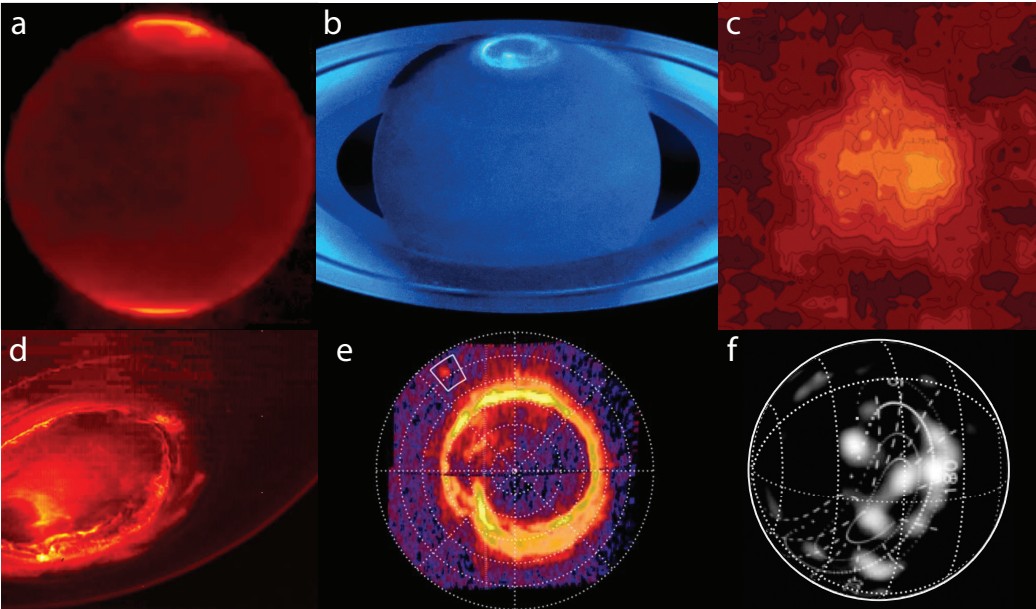

**Figure 1.** Upper-atmospheric emissions of Giant Planets. Earth-based telescope observations are shown in (**a**–**c**) while spacecraft observations taken near each planet are shown in (**d**–**f**). (**a**) IR view of Jupiter at 3.4 μm, capturing emissions from the major upper-atmospheric ion $H_3^+$. This image was taken by NSFCam on the NASA Infrared Telescope Facility (IRTF) [11]. (**b**) Saturn in UV, capturing emissions from upper-atmospheric hydrogen, taken by the Hubble Space Telescope (HST) in 2018; additional credits NASA/ESA, NASA and L. Lamy (Observatoire de Paris). (**c**) Uranus by NSFCam/IRTF in 1998 capturing $H_3^+$ emissions at 3.5 μm [12]. (**d**) The first detailed view of Jupiter's southern $H_3^+$ aurorae at 3.45 μm in 2016, taken by the Juno spacecraft's JIRAM imager as reported by [13]. (**e**) A detailed view of Saturn in UV recorded by the Cassini spacecraft in 2008, showing the northern auroral oval. A boxed region denotes emissions at the magnetic footprint of Enceladus [14]. (**f**) A close-up view of Uranus in UV taken by the Voyager 2 spacecraft flyby in 1986 [9,15].

## 2. Introduction to Giant Planet Upper Atmospheres

Upper atmospheres are often broken down into two components: a thermosphere of neutral atoms and molecules and a co-located ionosphere composed of ions and electrons [16]. Figure 2 shows the major neutral and ion species, which make up the upper atmospheres of Jupiter, Saturn, Uranus and Neptune as functions of altitude above the 1-bar 'surface' of each planet. Molecular hydrogen ($H_2$) is the major neutral species at most altitudes. Atomic hydrogen (H) forms a shelf of high density peaking at around 500 km and becomes the dominant neutral above 3000 km at Jupiter and Saturn, while this occurs above 5000 km for Uranus and Neptune (not shown here) [17,18]. $H_2$, H atoms, helium (He) and methane ($CH_4$) separate according to their molecular and atomic masses, with heavier elements resting at the bottom of the upper atmosphere. Indeed, the differences in gravitational acceleration at each planet largely explains the altitudinal differences in density for each species in Figure 2. On Saturn, water becomes a significant neutral species between 0–500 km planet-wide owing to influx from outside the planet from the rings and Enceladus [19,20]. Hydrocarbons, hydrocarbon ions and water ions are present between 0–1000 km at all four planets but are not shown here for simplicity.

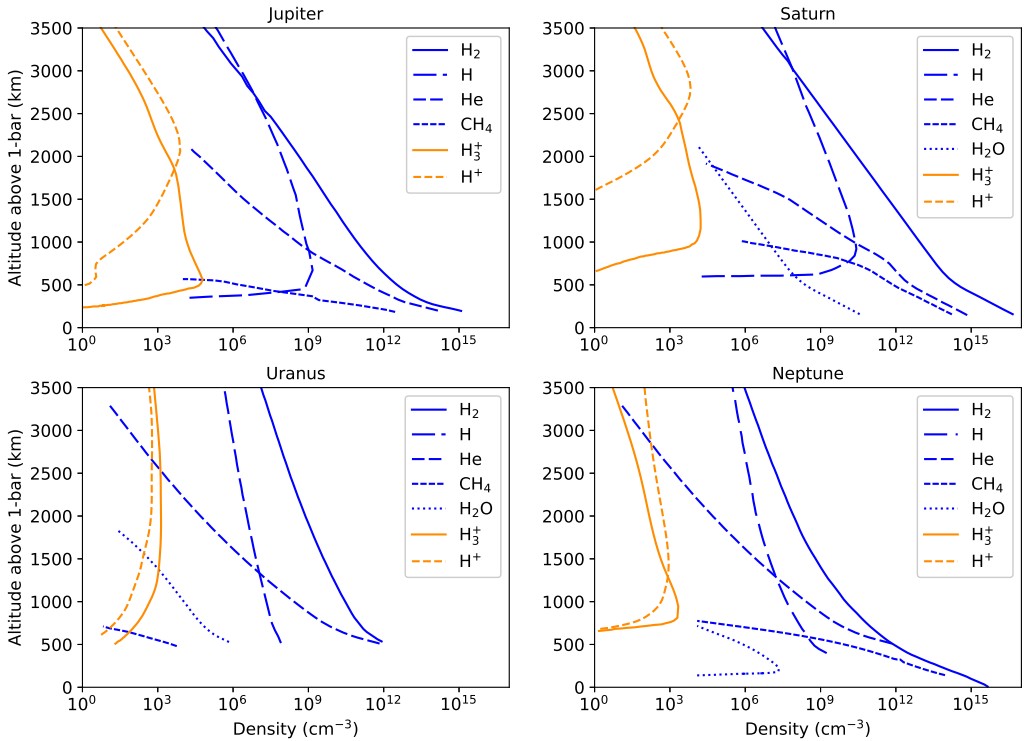

**Figure 2.** Number densities of major neutral and ion species are shown as a function of altitude above the 1-bar pressure surfaces of the four giant planets. The plots show modeled densities for Jupiter and Saturn [21], Uranus [17] and Neptune [18]. The solar zenith angle (where 90° corresponds to the Sun being directly overhead) varies in these models: 75° for Jupiter and Saturn, 15° for Uranus and 45° for Neptune, so care must be taken when comparing ion densities between the planets. In addition, Jupiter and Saturn ion densities are enhanced by charged particle precipitation associate with auroral mechanisms.

It is advantageous to the study of upper atmospheres that they are tenuous environments: with number densities ∼millions of times less than those of sea level air on Earth [21], they are uniquely sensitive to (and therefore easily reveal) physical and chemical processes the planet is subjected to. These outermost layers of planets are the first to experience influences from space such as electromagnetic radiation and the precipitation of matter from interplanetary space. Solar extreme ultraviolet radiation is absorbed in the thermospheres of giant planets, enough to raise upper atmospheric temperatures to between 130 and 200 K planet-wide [3], though temperatures are observed to be far hotter for reasons described in the next section. By far the most energetic upper-atmospheric process occurs in the magnetic polar regions. Here, aurorae emit powers ranging from tens of gigawatts (GW) at Neptune, hundreds of GW at Saturn and Uranus to tens of terawatts (TW) at Jupiter [5]. Heating associated with auroral activity, driven by ion-neutral collisions in the upper atmosphere, raises magnetic polar altitude-integrated temperatures to over 1000 K at Jupiter [22,23] and over 500 K at Saturn [24,25].

Modeled thermospheric temperatures as a function of altitude are shown in Figure 3. These temperatures rapidly increase with altitude in the thermosphere before becoming largely isothermal in the high altitude exospheres of each planet. The exosphere is the uppermost region of a planetary atmosphere and the region from which atmospheric escape occurs since the mean free path of the constituents here is greater than the scale height [26]. Thus, when provided with enough kinetic energy through collisions, exospheric neutrals and plasma can freely escape planetary gravity wells and enter space, which leads to a gradual transition from atmosphere to space. The thermal escape rate is directly proportional to mass, so lighter species such as atomic hydrogen escape more easily. If

the escaping species are also electrically charged, they instead become bound to magnetic field lines owing to Lorentz forces, causing them to oscillate along magnetic field lines between the magnetic poles or lost to the solar wind by magnetic reconnection in the tail of a planetary magnetosphere.

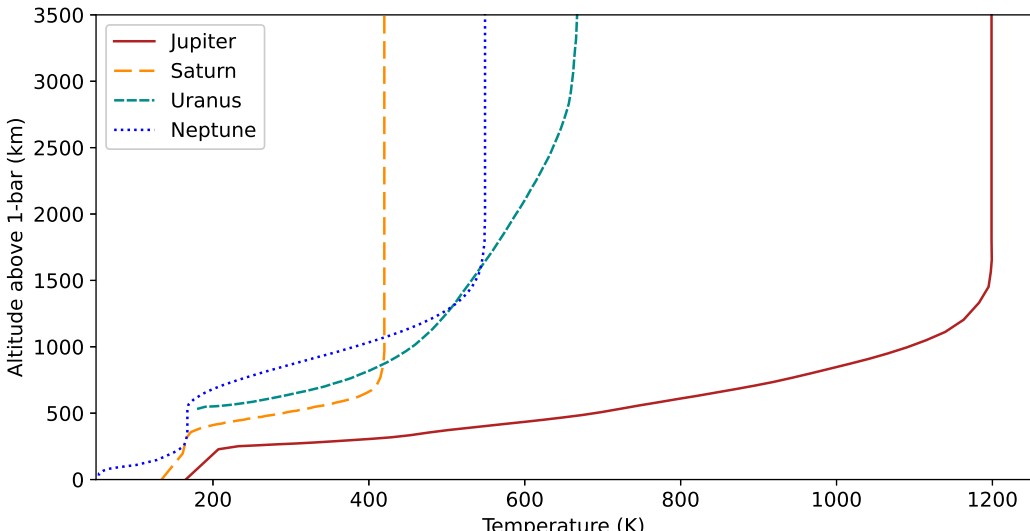

**Figure 3.** Models of neutral upper atmosphere temperatures for the giant planets Jupiter, Saturn [21], Uranus [17] and Neptune [18] as a function of altitude above the 1-bar pressure surfaces. The simulations of the upper atmospheres of Jupiter and Saturn include auroral heating. Temperatures in planetary upper atmospheres are highly variable temporally and spatially, so as with Figure 2, care must be taken when comparing temperatures between the planets, as we shall see in the proceeding sections. In other words, these values may be taken as broad averages.

## 3. Observing Giant Glanet Upper Atmospheres

$H_3^+$ is one of the main ions used to add to our understanding of upper atmospheres [27,28]. It is mainly produced in molecular hydrogen-rich environments by the reaction $H_2 + H_2^+ \rightarrow H_3^+ + H$ [29], therefore upper atmospheric $H_2^+$ production controls the amount of $H_3^+$. There are two main pathways by which $H_2^+$, and therefore $H_3^+$, is produced: ionization by solar extreme ultraviolet or by incident-charged particle precipitation. The former acts planet-wide, maximizing at the equator and diminishing at the poles owing to the angle of the Sun, whereas the latter dominates at the magnetic poles as a result of auroral mechanisms. $H_3^+$ ions are lost through recombination with electrons and charge exchange with neutrals [30], with the latter leading to the characteristic 'shelf' of ion density seen in Figure 2.

The $H_3^+$ ion has millions of rotational-vibrational (ro-vibrational) transition lines, the strongest of which are in the near-infrared between 2 to 4 μm. Spectroscopic analysis of $H_3^+$ emission lines can be used to determine the ion's temperature, density, total emission rate (also known as cooling rate) and line of sight velocity [31,32]. The measured intensity of $H_3^+$ correlates linearly with the ion's density and to the fourth power with the ion's temperature [2]. As $H_3^+$ is considered to be quasi-thermalized with its surroundings, its temperature is used to represent that of the entire upper atmosphere [21,25,33,34]. Strong methane absorption near 3.5 μm removes data-contaminating reflected sunlight at giant planets, allowing more $H_3^+$ emission lines to be detected across the planetary disk, including in the highly-reflective low latitude regions. The $H_3^+$ ion is therefore a key probe, which can be used to evaluate global thermal and chemical changes, as well as atmospheric circulation patterns in the upper atmospheres of giant planets.

Table 1 shows when the $H_3^+$ ion was discovered (or attempts were made) and investigated in terms of parameters in the upper atmospheres of giant planets. Aurorae are several times brighter than non-auroral regions, so $H_3^+$ was detected there first. Emissions

of $H_3^+$ were detected at Saturn and Uranus prior to deriving parameters (such as temperature), as the calculation of parameters requires the use of two or more strong emission lines. Ground-based observatories can conveniently observe the strongest $H_3^+$ emissions in atmospheric windows between 2–2.3 and 3.4–4 μm using high-altitude observatories, such as Maunakea in Hawaii, which stands at over 4000 m above sea level. Thermal emissions from Earth's atmosphere must be removed from ground-based data, requiring a telescope to nod between and record a given target and the nearby sky throughout an observing night. To determine the effect of Earth's atmosphere on infrared radiation as a function of wavelength, we must first measure the spectrum of a standard star. This allows us to identify the wavelengths of infrared light that are absorbed by the atmosphere, and to apply corrections to account for it. Space-based platforms, such as the James Webb Space Telescope, are capable of observing $H_3^+$ without a nodding pattern.

**Table 1.** Dates $H_3^+$ was first investigated in the upper atmospheres of giant planets.

| Planet | Auroral $H_3^+$ | | Non-Auroral $H_3^+$ | |
|---|---|---|---|---|
| | Emissions | Parameters | Emissions | Parameters |
| Jupiter | September 1988 [35] | September 1988 [35] | April 1992 [36] | April 1992 [36] |
| Saturn | July 1992 [37] | July 1992 [37] | Oct. 1998 [7] | April 2011 [38] |
| Uranus * | April 1992 [39] | April 1992 [39] | April 1993 [8] | June 1995 [8] |
| Neptune | Non-detection [40–42] | Non-detection [40–42] | Non-detection [40–42] | Non-detection [40–42] |

* Uranus is poorly resolved spatially from Earth, so auroral and non-auroral $H_3^+$ emissions are necessarily combined in $H_3^+$ observations so far. Here, we have simply taken the first detection of spatial $H_3^+$ variability as the first non-auroral detection.

Jupiter's neutral hydrogen ultraviolet emissions were first detected by a sounding rocket experiment over 50 years ago [43]. It is essential to go beyond Earth's UV-absorbing ozone layer (15–30 km altitude) in order to observe UV targets, so space-based platforms such as the HST and UV instruments aboard space probes (e.g., Juno at Jupiter, Cassini at Saturn) are some of the major tools used. UV emissions mainly emanate from the aurorae of Jupiter and Saturn following electron impact excitation in the form of H Lyman-alpha at 121.6 nm and $H_2$ Lyman and Werner band emissions between 90–170 nm [2]. The penetration depth of electrons into an atmosphere, which determines the altitude of UV emission, is governed by the kinetic energy of the precipitating electrons. On Jupiter, electrons with kinetic energies of 0.1 keV to 100 keV (corresponding to velocities of 0.02 to 0.55 c) can reach altitudes of 1800 km to 400 km, respectively [21]. Due to shorter wavelengths of UV light being absorbed by hydrocarbons more strongly than longer wavelengths, the ratio of UV emissions at long and short wavelengths can be used to infer the altitude where UV emissions are coming from, hence the penetration depth [44]. Larger 'color ratios' correspond to deeper penetration since shorter wavelength UV emissions were preferentially removed by a longer altitudinal column of hydrocarbons before arriving at the detector.

Ultraviolet emissions occur when electrons in hydrogen relax to lower electronic states, so when particle precipitation ceases, the emissions end within a fraction of a second [2]. Conversely, $H_3^+$ emits thermally, with energy stored in ro-vibrational states emitted over the lifetime of $H_3^+$, which lasts on the order of minutes to hours [11,45]. Observations of $H_3^+$ are thus natural long-exposures of events, whereas UV emissions are considered 'prompt', yielding an instantaneous view of events. Infrared quadrupole transitions of molecular hydrogen have also been detected in auroral regions of Jupiter, near 2 μm [46]. Infrared $H_2$ emissions are over an order of magnitude weaker than the strongest, Q- and R-branch $H_3^+$

emissions between 3–4 microns [47], so observations of the latter are more common in the literature. Auroral X-ray emissions have been detected at Jupiter and Uranus, resulting from the precipitation of MeV-energy high-charge state oxygen, sulfur or carbon ions, but to-date no X-rays associated with Saturn's aurora have been detected [48–50]. X-ray photons are absorbed by Earth's atmosphere and so must be observed by space-based telescopes, such as Chandra X-ray observatory and X-ray Multi-Mirror Mission (XMM Newton).

Solar and stellar UV emissions are diminished at specific wavelengths as they pass through planetary atmospheres, which causes $H_2$ and H to become ionized in the process. These attenuated emissions can then be intercepted and studied by UV-capable spacecraft, allowing for the retrieval of upper-atmospheric $H_2$ and H densities [51]. Partial pressures for each molecule and atom can be deduced from these densities, allowing neutral temperatures to be found using the ideal gas law. Radio occultations operate under a similar principle, but instead quantify the attenuation of a spacecraft's own radio emissions at radio telescopes back on Earth, which allows for electron densities to be determined as a function of altitude [52]. Radio waves generated by fast electrons traveling along magnetic field lines above the upper atmosphere have been observed at Saturn by the Cassini spacecraft (mission ended in 2017) and at Jupiter by the Juno spacecraft (still operational), offering insights into the energies of charged particles being accelerated into the planet [53,54]. Radio data are regularly used in conjunction with data taken simultaneously from other instruments, such as ultraviolet, particle and field detectors, as spacecraft closely pass over auroral regions [55,56].

## 4. Giant Planet Aurorae

Aurorae were the first feature observed in giant planet upper atmospheres owing to their bright emissions. They are produced by momentum and energy exchange between planetary ionospheres and plasma that is trapped in the magnetosphere outside of the planet in the equatorial plane. The entrained plasma at Jupiter is sourced of eruptions from the volcanic moon Io, whereas the dominant sources of plasma internal to the magnetospheres of Saturn, Uranus and Neptune are their icy moons and rings [18,57,58]. Large-scale current systems are set-up along the magnetic field lines that connect the two regions, leading the acceleration of charged particles into and across the polar region [59,60]. The resulting electron precipitation occurs near the boundary between open magnetic fields, which connect to the solar wind, and closed field lines, which connect to the planet at both ends. At Jupiter, the main auroral emissions form a kidney bean shape in the northern hemisphere, which is off-center to the Jovigraphic north pole, while the southern emissions form a circular shape, which is centered on the southern pole, and can be seen in Figure 1a,d. These and the aurorae of other planets are generally referred to as 'main auroral ovals', distinguishing them from auroral activity inside and around the main emissions, which have a myriad of driving mechanisms.

$H_3^+$ was discovered for the first time astronomically in 1989 during a search for Jovian polar $H_2$ quadrupole emissions [35]. Owing to larger telescopes, more advanced instrumentation, increases in computing power and improved models of the $H_3^+$ ion, we have now moved to single-point measurements to making maps of Jupiter's auroral parameters with unprecedented clarity. Figure 4 demonstrates this progress with 8.2 m Very Large Telescope (VLT) observations taken in 2012, which showed Jupiter's northern auroral parameters measured using $H_3^+$ as a 'probe' [23]. Jupiter's main auroral oval is seen clearly in terms of emission and column density as this is the site of particle precipitation, hence $H_3^+$ production and emissions. Temperatures are instead raised by complex electric currents, which heat the atmosphere by flowing along the main oval and in between, while line of sight velocities reveal ionospheric winds and circulation. In the study, temperature maps of the northern aurora recorded at different times of night showed that regions of aurora change temperature by several tens of Kelvin within hours, indicating that the Jovian auroral heating is highly variable and has complex small-scale driving mechanisms.

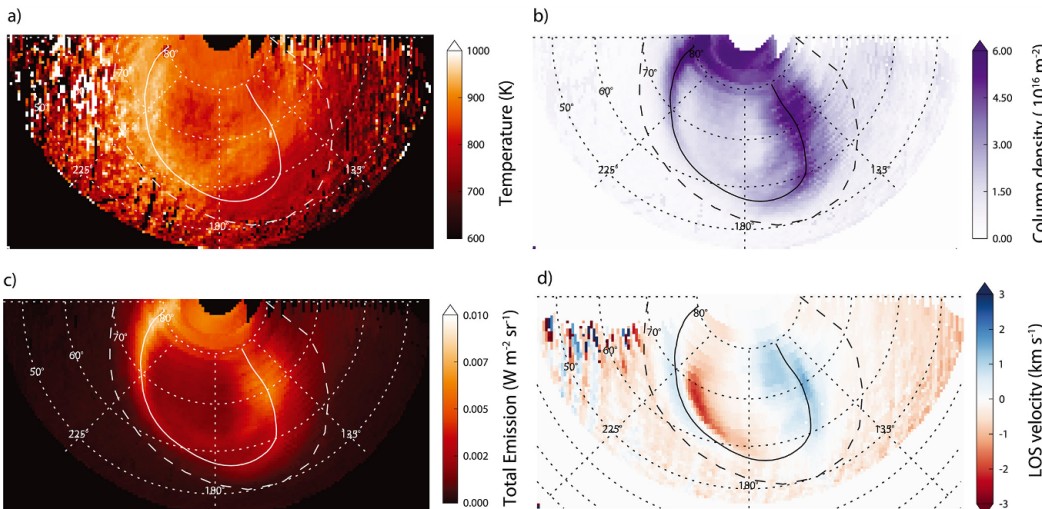

**Figure 4.** Polar projection of Jupiter's northern hemisphere showing H$_3^+$ parameters on 31 December 2012. Longitudes are magnetic System III, and the latitudes are planetocentric. The white line denotes the position of the main auroral oval [61]. The dashed white line indicates the magnetic footprint of Io [62]. Panels highlight the key H$_3^+$ parameters (**a**) temperature, (**b**) average column density, (**c**) average total emission, and (**d**) the line-of-sight velocity. Figure taken from [23].

Since the 1990s, observational studies of Jovian ultraviolet auroral have mainly been performed by the Hubble Space Telescope (HST) with either stand-alone studies or in support of other missions, such as the Juno spacecraft at Jupiter [2,63,64]. As Jupiter's axial tilt is only 3°, it is only possible for Earth-based platforms to see the edges of the southern aurora, but the northern aurora's kidney bean-shaped main oval, being off-center from the pole, can be viewed more clearly, when it rotates into view. Recently, and for the first time, Juno observed a type of auroral storm at Jupiter in which UV emissions become extremely powerful. For decades, these storms appeared to enter the field of view in HST observations on the dawn side of the planet, leading to them being called 'dawn storms'. However, Juno is in a polar orbit and was able to witness the development of a storm near the night-side, shortly before being captured by HST as a dawn storm [65]. These observations showed that dawn storms might be morphologically similar to 'sub storms' observed at Earth, which are caused by explosive reconfiguration of the tails of the magnetic field. Despite the fundamental differences between the magnetospheres of Jupiter and Earth, both experience forcing that leads to similar auroral activities, which are shown in Figure 5.

Long-term observations have demonstrated that Jovian UV auroral emissions, despite being driven chiefly by internal processes, change in response to the solar wind [66]. Long-term observations performed by the Japanese Aeronautics and Space Administration (JAXA) Hisaki space telescope, which has been in orbit around Earth since 2013 and continues to operate, despite having a planned mission lifetime of 1 year [67]. The Hisaki telescope features a unique dumbbell-shaped instrument, which allows it to simultaneously observe and compare the emissions from the torus of plasma surrounding Jupiter (on both sides of the dumbbell) with the Jovian UV aurora (on the 'handle' of the dumbbell). Hisaki observations have revealed that intense auroral bursts of emissions, which last under 10 h, coincide with variations in the parameters of the solar wind, such as its dynamic pressure, which compresses the Jovian magnetosphere. Furthermore, these bursts were found to be more common during times when volcanism on Io was more active, as the additional output from the moon increased the density of plasma torus [68]. However, some auroral brightening events occur independently of either Io or solar wind changes [69].

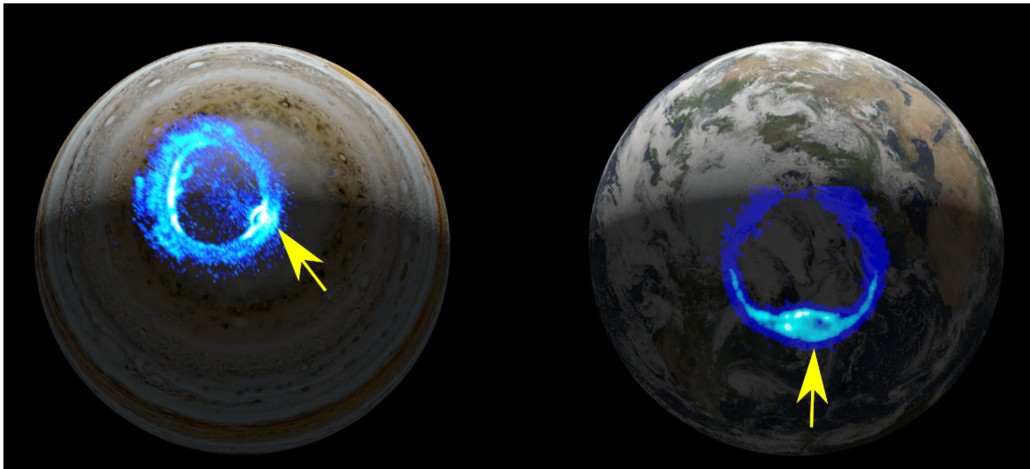

**Figure 5.** Polar projection of Jupiter's southern and Earth's northern hemisphere showing UV auroral emissions, placed over simulated visible light images of each planet (not to scale). The left image shows imagery from the Juno spacecraft's UVS (Ultraviolet Spectrograph) instrument. The image on the right corresponds to Earth's aurora as seen from the Wide-field Imaging Camera (WIC) on board NASA's Imager for Magnetopause-to-Aurora Global Exploration (IMAGE) spacecraft. The yellow arrows indicate auroral 'dawn storms' on both planets. Figure from the associated press release of [65].

Saturn's auroral main ovals are almost circular in both hemispheres and centered essentially perfectly over the poles. Statistical studies using Cassini spacecraft observations show that the main oval is slightly equatorward of the open-closed magnetic field line boundary near 76.5° (planetocentric) co-latitude in the north and 75° co-latitude in the south, as seen in Figure 6 [70,71]. This remarkable symmetry is due to Saturn's (dipole) magnetic field being almost completely aligned with the rotational axis of the planet to within less than 0.1° [72]. Despite these apparent similarities, the study shown in Figure 6 found that the southern main oval was a factor of 1.3 more intense than the northern main oval, while northern polar emissions (between 0–10°) were over twice as intense as those seen in the south.

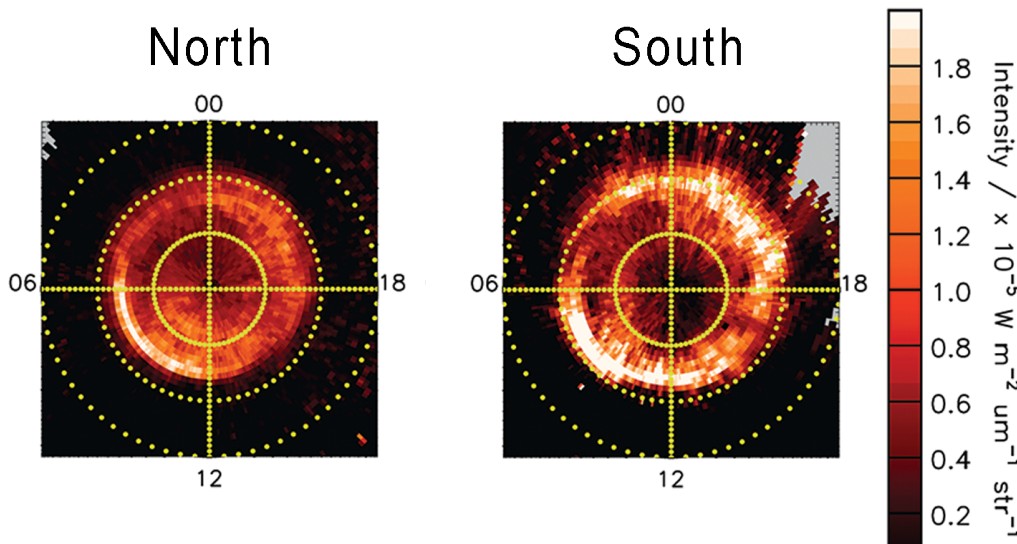

**Figure 6.** Polar projections of Saturn's infrared aurora by the Cassini spacecraft's Visual Infrared Mapping instrument. Two infrared channels were averaged together to produce the data: $3.532 \pm 0.0161$ µm and $3.667 \pm 0.0184$ µm, such that the majority of emissions are from $H_3^+$. Modified figure from [71].

Auroral temperature studies at Saturn were initially confined to time-averaged values over a night of observations due to colder temperatures relative to Jupiter, for example, values of 450 K were recorded by the 3.8 m UK InfraRed Telescope (UKIRT) in 1999 and 2004 [73]. Ground-based studies could only sample one hemisphere at a time owing to Saturn's 26.7° axial tilt, which puts the aurorae out of view of Earth-based platforms, while the Cassini spacecraft was well placed to view a single hemisphere during the closest approach in its orbits. Studies with the Cassini spacecraft found temperatures of 560–620 ± 30 K in 2007 and 440 ± 50 K in 2008 [41,74]. Using the large, ground-based 10 m Keck telescope in 2011, temperatures were reported in both hemispheres simultaneously for the first time, which allowed comparisons to be drawn between the two auroral regions under similar conditions; an important consideration given the several tens of Kelvin variability seen within hours at Jupiter [23].

The signal to noise ratio of Keck observations was sufficiently high that 10 measurements of temperatures could be taken within a single night, yielding one measurement every 15-min [24]. In the new Keck study, northern auroral temperatures were 527 ± 18 K, while southern values were 583 ± 13 K. This asymmetry is likely due to an inversely proportional relationship between the total heating rate and magnetic field strength, which is stronger in the north. As a consequence, the size of the auroral oval is larger in the south and therefore the total heating delivered is greater [75]. Saturn's equinox in 2009 also gave a rare chance for HST to view UV emissions from both hemispheres simultaneously, as seen in Figure 7 [76]. The authors found that the UV power emitted is proportional to the magnetic field strength, as the northern total UV power was 17% larger than the south. This result is opposite to the total heating case, as UV emissions were thought to be driven by magnetic field-aligned currents in this case [76].

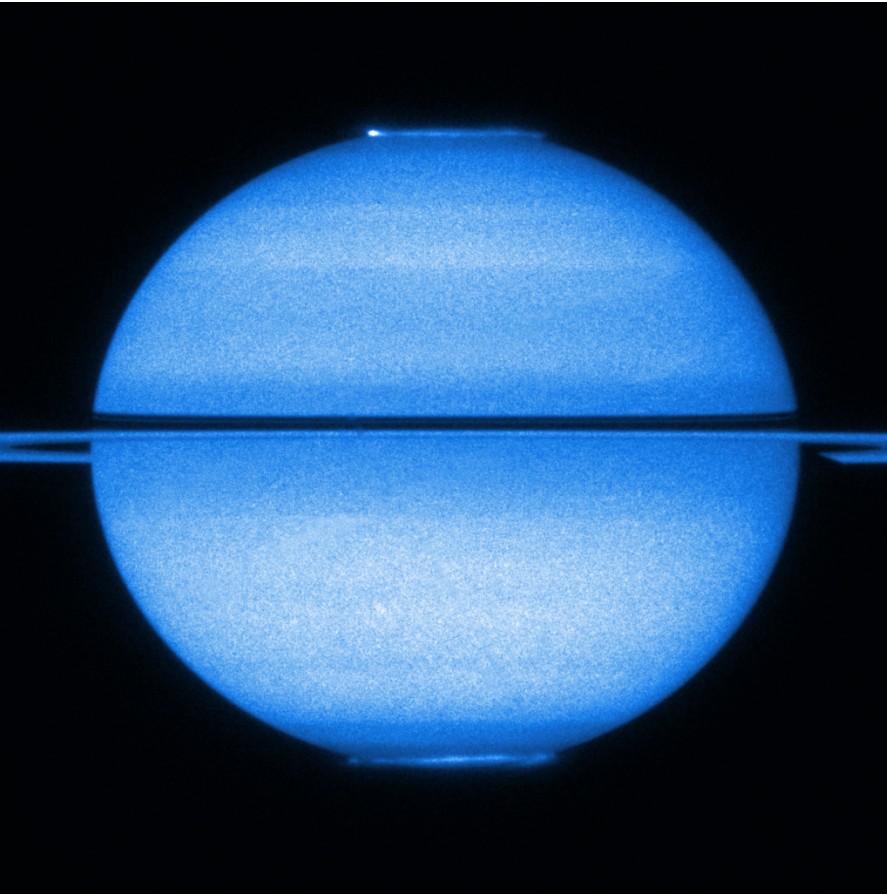

**Figure 7.** Saturn at equinox, taken by the Hubble Space Telescope in 2009. Saturn's rings are seen edge-on and both poles are in view, offering a simultaneous view of both aurorae. Figure from the associated press release of [76]. Credit: NASA, ESA and Jonathan Nichols (University of Leicester).

Ion wind velocities within the polar region of Saturn's aurorae show a significant sub-corotation, with the magnetospheric breakdown in corotation mapping into the ionosphere [77]. There is some evidence of localized ion wind flows across the auroral region [78,79], but these do not appear to align directly with the main auroral emissions, suggesting that we still do not have a clear picture of the closing currents that are associated with this main auroral oval. However, it has long been known that, despite the direct alignment between the magnetic and rotational pole, there remains a notable offset in the auroral emission [80], which ultimately results in an apparently variable rotation rate for each hemisphere. More recent measurements above the auroral region of Saturn have revealed that the main auroral emission consists of two separate components—the first, an axi-symmetric main emission perhaps associated with the outer magnetosphere, but also a second similarly powered emission that is directly associated with a twin-cell current system that rotates with the planet [81]. The origin for this rotating current system has been the source of significant speculation within the research community [82]. Utilizing a sequence of Keck observations, [83] measured the ion wind flows in phase with the variable rotation rate of the radio aurora, and were able to detect the clear signature of a twin-cell vortex across the northern auroral region. The direction of these winds showed that the ions, and so the entire auroral current system, is being driven by the neutral atmosphere, rather than the magnetosphere, providing the first evidence of an atmospherically generated aurora in the solar system. This theory, first discussed by [84], highlights the importance of complexity with the neutral upper atmospheres of giant planets to the dynamics seen in both their auroral systems and magnetospheric environments.

## 5. Global Upper Atmospheric Weather and Circulation

Earth's upper atmospheric temperatures are over 800 K during quiet auroral conditions [85], and these temperatures are sufficiently explained by solar heating alone. Sunlight at the orbit of Jupiter (for example) is just 4% as intense, but surprisingly, equatorial temperatures are similar to those seen at Earth (700 K). In fact, the upper atmospheres of all giant planets are hundreds of degrees hotter than models predict outside of their auroral regions, with sunlight only providing enough energy to raise temperatures to about 200 K [3,86]. This large discrepancy, or 'energy crisis', has existed for nearly 50 years [87], demonstrating that basic physical processes have not yet been understood. The two main candidates for heating are the redistribution of auroral energy from the poles and acoustic or gravity wave heating from the lower atmosphere. Models of global circulation demonstrate that auroral heat cannot be effectively distributed at Saturn or Jupiter [88,89], prompting a search for alternative sources of heating, such as gravity of acoustic (sound) waves from below. Both types of wave would essentially dissipate their momentum as they 'break' in the thermosphere, heating the region much like ocean waves crashing on a beach. Gravity waves have been found to heat the upper atmosphere by several Kelvin, or even cool it [90,91]. Acoustic wave heating on the other hand, has been shown to supply significant heating, with storms being a prime candidate for their generation [92]. Indeed, observations found such heating above Jupiter's Great Red Spot (GRS), but more recent observations show no significant heating above the GRS, potentially indicating such a mechanism is time-variable [93,94].

The first maps of Jovian upper-atmospheric temperatures, which can be used to discern between the missing heat sources, were produced using data from UKIRT [47], but spatial resolutions were so low that only about two pixels covered the 45–90° latitude region in each hemisphere. Heat redistribution from the polar-auroral region to the equator may have been present, though equatorial temperatures were also found to be similar to auroral values. These findings would indicate that auroral heat re-distribution was possible and also that a heat source is active at low latitudes. A later study tracked short-term $H_3^+$ temperature changes in Jupiter's auroral region, finding that local cooling by radiation to space by $H_3^+$ was too inefficient to explain atmospheric cooling: this was compelling evidence that heat is transported from the auroral/polar regions by winds [22]. Later, however, global circulation modeling could not demonstrate the distribution of

heat globally from the aurorae. The high-temperature upper atmosphere was found to be confined to the auroral/polar regions at Jupiter by strong zonal winds [89], while attempts to circulate Saturn's auroral energy globally actually led to a net cooling effect at lower latitudes [88].

Observational breakthroughs on this problem have occurred recently. At Saturn, UV stellar and solar occultation measurements taken by the Cassini spacecraft between 2005 and 2015 (combined with older Voyager 2 occultation measurements) show a temperature decrease from the poles to near the equator [86,95]. These measurements were made at a variety of local times over a third of a Saturnian year, such that they include a range of different solar illumination angles, but they provided early evidence of a temperature gradient between the aurorae and the equator at Saturn. With the end of the Cassini mission in 2017, 33 more stellar and solar occultations were added to the over 50 accrued values recorded in the previous decades. From these, pole-to-pole latitude–altitude temperature and density profiles were produced, which again, found a negative temperature gradient between the auroral region and low latitudes [96]. Westward zonal winds, which normally act to confine the relatively hot polar upper atmosphere, were also derived from these measurements. These were found to be 500–800 ms$^{-1}$ and exist over 30 degrees of latitude instead of the 1100–1400 ms$^{-1}$ over 10 degrees of latitude predicted by the global circulation model used in the study. These slower and broader flows are thought to allow equatorward winds to ferry hot atmosphere to lower latitudes in both hemispheres.

Soon after, at Jupiter, two $H_3^+$ temperature maps in terms of longitude and latitude were produced using ground-based Keck telescope data from two single nights in 2016 and 2017, comprising several thousands of data points each. These were produced solely from data at local noon, so each longitude and latitude on the map was under the same solar conditions. Both maps revealed negative temperature gradients between the auroral region and equator, with temperature monotonically falling from the poles to the equator and adjacent latitudes [94]. Coriolis forces and related effects were observably overcome at Jupiter, such that the aurorae are permitted to circulate their heat planet-wide. At Jupiter and Saturn, we thus have multiple complementary views demonstrating negative temperature gradients from the auroral regions to the equator in terms of longitude, latitude and altitude; these 3-dimensional views are shown together in Figure 8. Combined with earlier evidence that the aurorae are cooling so fast that their heat must be being transported away at Jupiter [22], there is now much evidence that the aurorae are the main source of the high temperatures observed globally at Jupiter and Saturn.

If heat is mechanically transferred from the aurorae by winds, how are the confining, fast zonal winds presented in global circulation models observably overcome? New evidence suggests that zonal winds are slowed by vertically-propagating atmospheric waves, which provide a friction, or dampening, to these flows. The final orbits of Cassini took the spacecraft through the planetary upper atmosphere, allowing for in situ measurements of neutral densities [97]. Atmospheric waves were discovered for the first time in these close passes, with vertical wavelengths on the order of 100–200 km and density variations of approximately 10%. The inferred wave-damping was expected to enhance eddy friction within the thermosphere, providing the necessary 'obstacle' to dissipate the momentum in the zonal winds. The authors concluded that under this scenario, equatorward winds are efficient to transport energy, leading to wind-driven global redistribution of energy at Saturn. Assuming such waves are common to all giant planets, the friction they provide may be the reason auroral energy is able to escape from the magnetic polar regions of Jupiter, Uranus and Neptune.

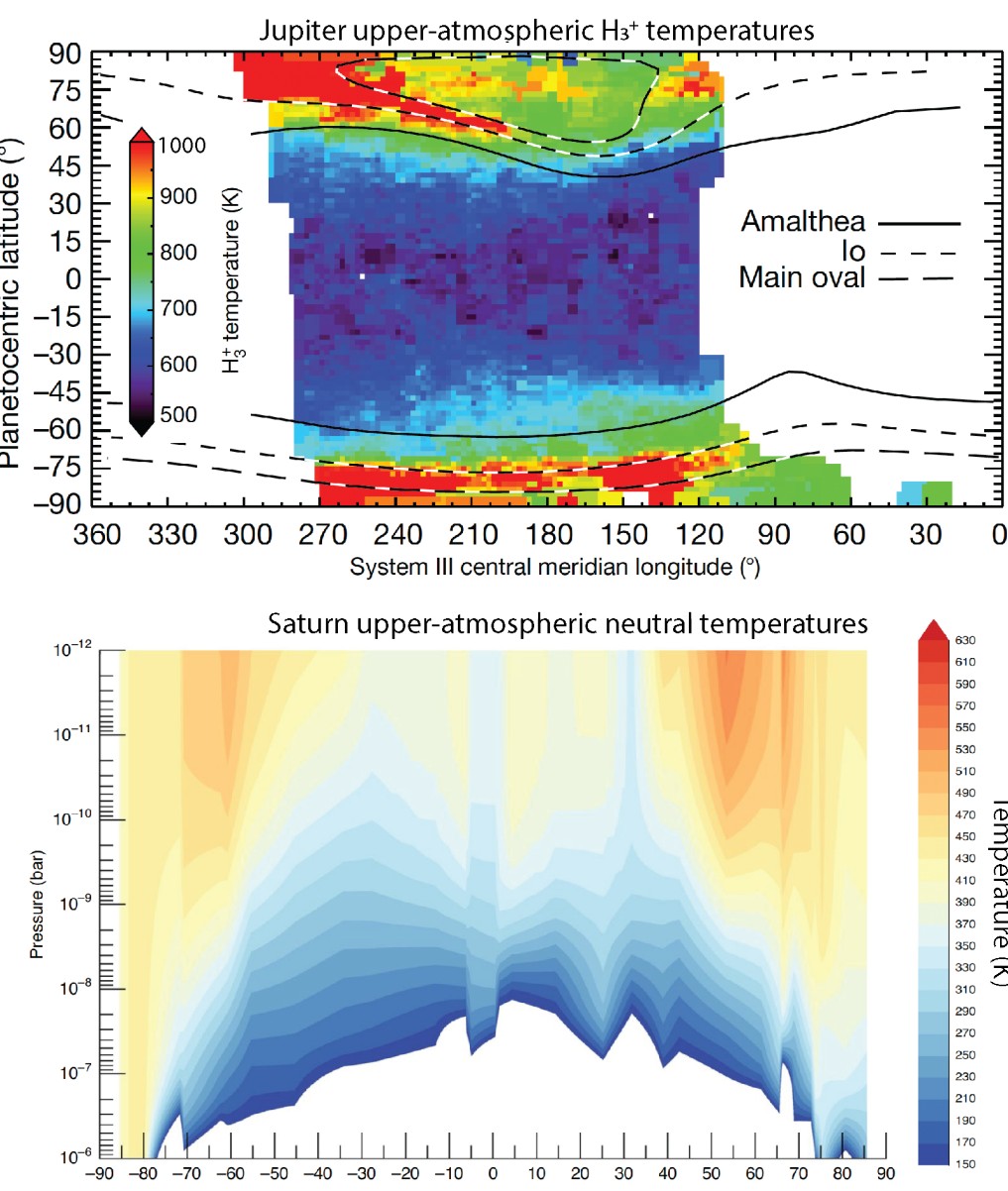

**Figure 8.** Observed upper-atmospheric temperatures at Jupiter and Saturn. (**Top**): equirectangular projection of Jupiter's altitudinally column-integrated $H_3^+$ temperature as a function of central meridian longitude (Jovian system III) and planetocentric latitude, adapted from [94]. Long black-and-white dashed lines show Jupiter's main auroral oval, short black-and-white dashed lines correspond to the magnetic footprint of Io, and the single thick black line corresponds to the magnetic footprint of Amalthea. Temperatures have uncertainties below 5%. (**Bottom**): Saturn's meridional temperature profile as a function of altitude and pressure (altitude) adapted from [96].

$H_3^+$ temperatures on Uranus were derived to be $740 \pm 25$ K when the ion was discovered in the planet's upper atmosphere in 1993 [39]. Following this, decades of ground-based measurements were recorded, allowing us to see a long-term temperature decrease unlike anything seen at Jupiter or Saturn [10,15,42,98]. From 1992 to 2018, the temperature of Uranus' upper atmosphere has linearly decreased by 8 K/year, from over 700 K to below 500 K. A year on Uranus lasts 84 Earth-years, so 1 season lasts 21 Earth-years. The long-term trend in temperature on Uranus is similar in scale to the length of its season, and given that the planet's axial tilt is 98°, some form of seasonally-driven change is thought to be a likely cause. During solstice at Uranus, one hemisphere is perpetually lit by the Sun while the other is in constant darkness (for 17 h and 12 min per Uranus day). During equinox,

conditions are the same as that experienced on Earth during the same conditions, with the entire planet receiving the same amount of sunlight hours per day. Two hot solstices and two cold equinoxes would be expected if solar heating was the major cause of heating in Uranus' upper atmosphere (known a 'geometric season'), but the temperature decline has continued for significantly longer than a Uranian season. Figure 9 shows that temperatures have continued to cool for 11 years beyond equinox, as the planet moves into southern summer. Even with an arbitrary shift in the seasonal cycle to account for thermal inertia, it is not possible to argue that the temperature decrease is caused by changing solar irradiance.

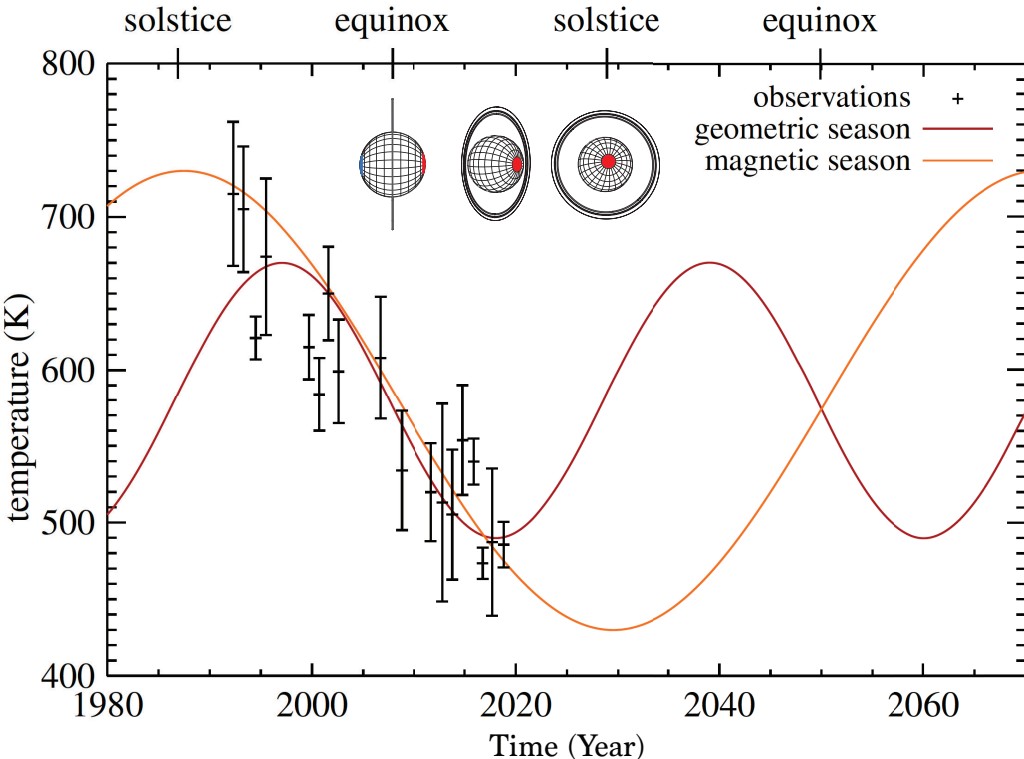

**Figure 9.** The long-term $H_3^+$ temperature trend measured at Uranus. Geometric and magnetic seasons are over-plotted and described in the main text. Inside the figure, a simple schematic view is provided to illustrate the changing configuration of Uranus over time, with a red cap added to a single hemisphere to guide the eye. Figure adapted from [99].

An alternative explanation relates to Uranus' magnetic field configuration. As well as Uranus being on its side, revolving around the solar system like a wheel, it has a highly inclined magnetic field tilted 60° with respect to its rotational axis. Furthermore, the dipole center of the magnetic field is 0.3 $R_U$ from the center of the planet (where 1 $R_U$ = 25,362 km). It is possible that one magnetic pole allows for more intense reconnection with the interplanetary magnetic fields embedded in the solar wind, thus enhancing auroral activity and upper-atmospheric heating [99]. Indeed, during conditions of equinox in 2007, the configuration of the magnetosphere was expected to inhibit reconnection [100]. The 'magnetic season' of Uranus, plotted on Figure 8, also illustrates that long-term observations of the planet are important for establishing which hypothesis is correct, or if some other mechanism is responsible.

Neptune's upper atmosphere has not been observed since the Voyager 2 spacecraft observed ultraviolet emissions in 1989 [101]. Upper limits have been established for $H_3^+$ temperature and density however, indicating either that the upper atmosphere has cooled by hundreds of Kelvin since Voyager 2 measurements (to 450 K or lower), or that $H_3^+$ densities are much lower than expected [15,18,42]. If lower temperatures are the cause of weak emissions, perhaps a magnetic season may be implicated at Neptune as well. If densities are lower than expected, it is likely that the planet is subjected to an influx of

neutral material, which is depleting $H_3^+$ significantly through charge–exchange reactions. There is a precedent for this, since a large abundance of CO [102] has previously been detected in the stratosphere, which the authors argued was sourced from a comet impact in the distance past. Ref. [18] produced a model Neptunian upper atmosphere simulating the effects of a CO influx, determining that it could explain the absence of $H_3^+$ emissions from a low density perspective, highlighting that out-gassing moon Triton could be an additional source of CO. However, the required influx of CO was larger than Triton can supply, so it is thought that the icy moon is either more active than previously thought or Neptune's rings are also contributing. A combination of lower temperature and lower density together could also result in the lack of $H_3^+$ emission [18].

## 6. Saturn's Ring Rain

Saturn's ring system is comprised of pieces of almost pure water ice orbiting the planet and ranging in size from below 0.01 cm to several metres, distributed in size following an inverse cubic power law so that most of ring material is small [103,104]. Submicrometre-sized icy grains pick up electrical charge by solar photoionization and, to a lesser extent, exposure to micrometeorite impacts, which are intense enough to form plasma clouds upon impact [105]. Following this, the main forces acting on the charged ring grains are gravity, which pulls the grains toward the planet along magnetic field lines, and a combination of centrifugal forces and magnetic mirror forces, which act to pull or confine the grains to the ring plane. The closest position to Saturn within the ring plane where these forces are able to balance for icy grains is determined to be 1.53 $R_S$, where 1 $R_S$ is Saturn's radius 60,268 km at the equator [106]. In this region and planetward of it, charged grains are unstable and rain into the planet. The steep density gradient observed between the B and C rings may indeed be the result of 'ring rain' electromagnetically eroding the rings away [107]. The Pioneer 11, Voyager and Cassini spacecrafts have found through radio occultation measurements that ionospheric electron densities were an order of magnitude lower than expected at mid- to low-latitudes [108–110]. An external influx of material such as water, perhaps from the rings, has long been thought to explain this, as $H^+$ (which recombines slowly with electrons) can be converted into water ions such as $H_3O^+$, which efficiently recombine with electrons, reducing them locally [30].

Ring rain as we know it today is a major process responsible for modifying the global chemistry in Saturn's upper atmosphere. Charged icy grains, which perhaps contain 1 charge per 1000 neutral molecules [105], fall in from the rings and sublimate in the relatively warm atmosphere at mid-latitudes, delivering packets of mainly-neutral molecules [111,112]. The first observations of this phenomenon came from Voyager 2 spacecraft green-filter images of Saturn, in which dark bands were found to encircle the planet at mid-latitudes. These dark bands indicated a reduction in sunlight being reflected from the planet to the spacecraft from high-altitude stratospheric hazes. The latitudes of these bands map out into space along magnetic field lines to Saturn's rings between the B and C rings, and also to the orbit of icy moon Enceladus, which is known to be out-gassing water inside the Saturnian magnetosphere [113]. Haze particles act as condensation nuclei to downward-diffusing water, so the dark bands are explained as the local removal of haze particles, as they attach to water, become heavy and sink to the atmosphere below. Indeed, photochemical models later demonstrated that stratospheric water locally depletes hydrocarbons [114]. HST observations were later performed in the UV, with hydrocarbon abundance examined in low spatial resolution at four latitudes. These observations showed depletions of hydrocarbons [115], which were consistent with an influx of neutrals (such as water) flowing from the rings to the atmosphere along magnetic field lines.

The 10-metre Keck telescope was used to observe Saturn's $H_3^+$ emissions from pole-to-pole in 2011 [19]. In those emissions, peaks in $H_3^+$ emissions were recorded simultaneously at 43° and 38° north and south, respectively. These regions share a common magnetic field line, which intersects the ring plane at approximately 1.53 $R_S$. The latitudinal offset of this feature in each hemisphere is a result of Saturn's (approximately dipolar) magnetic

field being centered slightly north of the planet's center. This asymmetry helped to reveal that the features were magnetically conjugate and less likely to be some other weather-related phenomenon unique to the mid-latitude upper atmosphere in both hemispheres. Subsequent modeling demonstrated that the increase in $H_3^+$ emissions was caused by an increase in $H_3^+$ density [30], indirectly driven by water ion destruction of electrons. It was found that when a small amount of water (or other neutral) flows into the planet, $H_3^+$ densities increase as electrons (which are a loss pathway for $H_3^+$) are locally removed. A large influx of water, on the other hand, can cause the loss rate of $H_3^+$ to increase as charge-exchange with water products starts to overtake the enhancement in $H_3^+$ by the reduction in electron density.

The signature of ring rain in $H_3^+$ emissions were re-detected in Keck telescope observations taken in 2013, but the emissions were a factor of four lower than in 2011 [116], likely because the upper atmosphere cooled by approximately 100 K and the emissions are directly proportional to temperature [86]. Despite the lower signal to noise ratio, however, the relative brightness between bright and dim features in the $H_3^+$ emissions was greater in 2013, potentially indicating that the influx of ring material had increased. Indeed, as the rings faced the Sun more directly in 2013, the production of charged icy grains ought to be larger, and hence, the precipitation to the planet greater [116]. In 2017, the Cassini spacecraft flew between the planet and rings and detected the presence of grains tens of nanometers in size near the ring plane and at mid-latitudes in both hemispheres; the latter a spectacular confirmation of the ring rain process [117]. The grains associated with the ring plane are associated with a newly-discovered equatorial influx of material in which grains of the innermost rings of Saturn are de-orbited upon colliding with the upper atmosphere (known as collisional drag) [118].

Estimating the mass loss of the rings is important for determining the age, total lifetime and evolution of the rings, which may have existed between 4.4 million and 4.5 billion years [105,107,119], an enormous uncertainty. Ring rain alone, it was determined, deposits between 432 and 2870 kg s$^{-1}$ of material into Saturn which, if constant, yields a future ring system lifetime of 168–1110 million years (ranges due to uncertainties) [30,38]. In approximate agreement, an estimated 100–370 kg s$^{-1}$ was found to be deposited at mid-latitudes by the Cassini spacecraft; the difference is possibly a result of the instrumentation being sensitive to only a subset of all ring grain sizes [117]. The equatorial influx driven by collision drag was determined to be much higher at 4800–45,000 kg s$^{-1}$, with the range of values encompassing variations in the amount of influx as a function of longitude. When combining both types of ring decay and assuming they are constant, the ring lifetime reduces much below 100 million years. These estimates assume that the rings are able to reorganize over time, with material slowly migrating from the outside in, and also that the rings will not entirely be removed. Saturn's rings are large compared to those of Jupiter, Uranus and Neptune, but given the existence of such erosion mechanisms, which may be universal for any planet with a magnetic field, this may not have always been the case.

## 7. Conclusions and Future

This review has discussed how we have obtained our knowledge of giant planet upper atmospheres using remote sensing measurements on Earth and in space. Much of what we have considered regards solar/stellar/radio occultation measurements, infrared spectroscopy of $H_3^+$ and ultraviolet emissions from $H_2$ and H. These studies have allowed us to begin to understand how energy balance and chemistry change in the upper atmosphere as a function of longitude, latitude, altitude, local time, season and under varying solar wind conditions. Despite significant progress in this field, many open questions remain and we highlight some of the major ones below and offer potential solutions:

- Are the auroras the main cause of planet-wide heating? Additional global temperature maps recorded over several years could confirm whether or not this is the case by giving a definitive view of Jupiter's upper-atmospheric climate and reveal the contributions from other heat sources. While global maps of temperature have been used to

infer transport through the study of temperature gradients, direct velocity measurements of the equatorward winds could remove much of the remaining doubt about this heat-circulation mechanism. Understanding of global energy balance will also be improved by measuring the magnitude, location and time-variability of heating delivered by other sources, such as from acoustic waves of storms below (e.g., the GRS at Jupiter).

- Do auroras redistribute heat at Uranus, Neptune and exoplanets, as they appear to do at Jupiter and Saturn? Observations of Uranus and Neptune at high spatial resolution can answer these questions, although Neptune's upper atmosphere is (so far) impossible to observe from the vicinity of Earth. Measurements of exoplanetary $H_3^+$ could be used to determine if giant planet upper-atmospheric temperatures are chiefly driven by mechanisms local to the planet, as opposed to stellar heating, if they can be shown to have anomalously high temperatures despite being separated a great distance from their parent star.

- Why do the temperatures of the upper atmospheres of Saturn and Uranus change slowly over long time scales (several years to decades)? Yearly observations taken over decades are important for understanding the apparent seasonal trends in temperature seen at these worlds. Global circulation models can also be improved for all planets by including new physics (such as wave-damping [97]), which can then be used to make testable predictions about future temperatures.

- What is the seasonal variability of Saturn's ring rain and the equatorial influx? Determining the likely large variation of influx is essential for understanding the evolution and lifetime of the ring system. Ring grain charging by solar EUV increases when the rings are fully exposed to the Sun during solstice and decreases during equinox. Long-term observations can be used to derive the (likely) varying influx of material as a function of latitude and season. The equatorial influx may also vary over the long-term, but owing to the equatorial region being obscured by the rings this is only possible near solstice (from Earth-based platforms).

New and future observation platforms will allow us to study the properties of giant planet upper atmospheres in unprecedented spatial and spectral resolution with exquisite sensitivity. The James Webb Space Telescope (JWST) began operation in mid-2022, offering some of the most sensitive views of the universe yet with approximately 100 times more sensitivity than Keck. Figure 10 shows a composite image by JWST and its near-infrared camera (NIRCam; 0.6 to 5.0 μm) instrument [120], the clarity of which surprised the authors of this review and scores of planetary astronomers world-wide. In the image, aurorae can be seen above the planet itself, extended for thousands of kilometers. The Sun in this image is slightly to the left, illuminating the western (left) limb fully while leaving a small, slither of shadow on the eastern (right) limb. Above the shadow on the eastern limb, a thin crescent-shaped region of brightness can be seen, which cannot be reflected sunlight. While the cause is not yet published in detail, we speculate that these are the emissions or 'air glow' from warm $H_2$ and $H_3^+$ in the upper atmosphere, which emit within the 2.12 μm filter and sits at around several hundred kilometers above the planet.

Spectroscopic data from the near-infrared spectrograph (NIRSpec; 0.6 to 5.3 μm) [121] have already been obtained for Jupiter, with early examinations showing that $H_3^+$ parameters can be successfully derived at low-latitudes on Jupiter at the highest spatial resolutions ever achieved (private communication, Henrik Melin). JWST will thus allow for the exploration of planets in a similar manner to the next-generation of ground-based telescopes, but a key advantage of JWST is its ability to probe all wavelengths of light free from atmospheric absorption. At Saturn, emissions from water-based ion $H_3O^+$ are important for understanding upper-atmospheric chemistry and determining the rate of water influx into the planet, but most of the brightest emission lines from this ion (at 10–20 μm) are blocked from the ground [122]. JWST's mid-infrared instrument (MIRI; 4.9 to 28.1 μm) [123] and as such represents the current best chance of detecting the ion's presence.

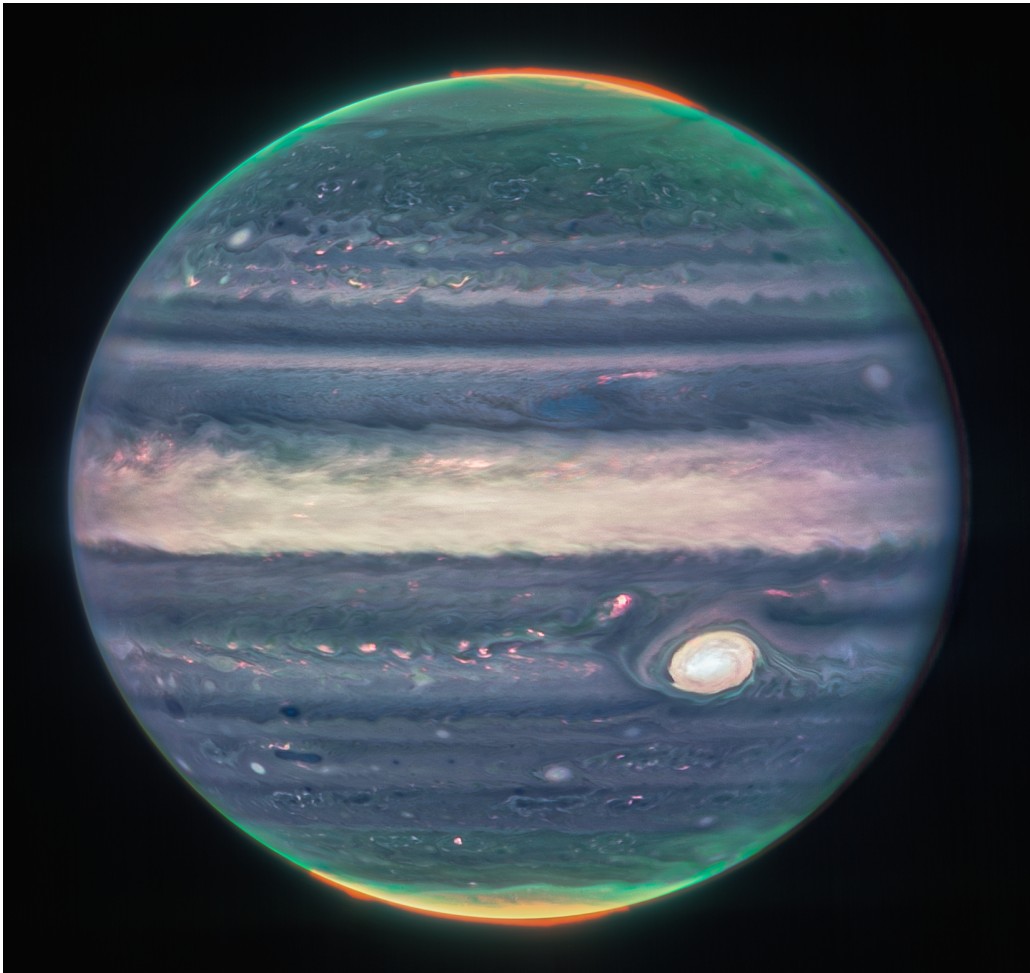

**Figure 10.** JWST composite image of Jupiter using three NIRCam filters—F360M (3.6 μm; mapped to red), F212N (2.12 μm; mapped to yellow-green), and F150W2 (1.5 μm; mapped to cyan). Each image was de-rotated before combining in order to account for the planet's rotation. Credit: NASA, ESA, CSA, Jupiter ERS Team; image processing by Judy Schmidt.

The current largest telescope being routinely used for astronomy of giant planet upper atmospheres, namely through the study of $H_3^+$ in the near infrared, is the 10 m Keck II telescope in Hawaii, with a collecting area of 76 m². Three of the largest observatories in history are currently under construction, which have effective apertures of 22 m (the Giant Magellan Telescope), 30 m (the Thirty Meter Telescope) and 39 m (the European Extremely Large Telescope), with planned completions aimed for the late 2020s. These telescopes use segmented mirrors like Keck rather than one monolithic piece, and have collecting areas of 368 m² (4.5 × Keck), 655 m² (8.6 × Keck) and 978 m² (12.9 × Keck), respectively. Lower diffraction limits will afford approximately an order of magnitude smaller (i.e., 'better') spatial resolution using adaptive optics, though each instrument will typically have smaller fields of view. These factors together mean that sub-planetary scale regions of the upper atmosphere, such as the aurorae, can be probed in unprecedented detail with low uncertainties.

Altitude profiles of $H_3^+$ temperature, density and emission above the limb will also be greatly improved, for example, at Jupiter, these parameters will be derived at ~10 s kilometres per pixel compared to the previous ~100 s kilometres per pixel. This information can be used to find the altitude of auroral energy deposition and characterize the vertically-propagating atmospheric waves, which are thought to play a pivotal role in allowing auroral energy to circulate planet-wide. The auroral ovals of Uranus (see Figure 1) may be resolved routinely, allowing researchers to determine if the source of heating planet-wide

is the auroras. Finally, these new technologies renew hopes for finding $H_3^+$ emissions at Neptune, or at least help to determine new upper limits of astrophysical importance.

The Juno spacecraft is currently at Jupiter until at least 2025, NASA's Europa Clipper will be present in the system between 2030–2034 and the European Space Agency's Jupiter Icy Moon Explorer (JUICE) spacecraft will visit between 2031–2034, although only Juno will specifically study Jupiter itself. A mission to Saturn's moon Titan is slated to arrive in 2034, which would fly a robotic rotorcraft (essentially an automated drone probe) over the surface. The next mission dedicated to the study of a giant planet is likely to be the Uranus Orbiter and Probe, which could potentially arrive at the planet in 2044 for a 4.5-year-long mission. Uranus and Neptune have never been orbited by a spacecraft, having had only a single fly-by each, so much will be learned about ice giants from such a mission. Currently, there are no missions planned for the exploration of Neptune. Ground-based and space-based telescope platforms will monitor these worlds when they are without a companion spacecraft from Earth. In addition, and crucially, Earth-based platforms will also support these missions by providing additional context to their observations (and vice versa).

The study of giant planet upper atmospheres in the solar system has enabled us to understand how planets interact with their environments; particularly in terms of energy balance, chemistry and revealing of interesting phenomena. With the new astronomical instrumentation outlined above, our knowledge can be expanded further and farther by detecting exoplanetary upper atmospheres through their $H_3^+$ emissions, and examining planet–star interactions for the first time. Auroral $H_3^+$ emissions at Jupiter and Saturn near 3–4 μm are the dominant emission in this wavelength, often orders of magnitude brighter than other planetary features [124], so a positive detection of exo-$H_3^+$ would thus strongly imply ongoing auroral processes and indicate the presence of a planetary magnetic field. GJ 504b, a Jovian exoplanet resolved around a solar-type star, is an example candidate for such a search. As it is separated from its host star by 2.5 arcseconds, a spectrum with minimal stellar contamination may be obtained using either adaptive optics from the ground or JWST. Such a spectrum can be used to derive column-integrated densities and temperatures of $H_3^+$, parameters that have for decades given a fountain of insights about gas giant upper atmospheres in our solar system.

Over the last few decades, our technological ability to remotely understand the properties of giant planet upper atmospheres has increased dramatically by every metric. This upward trend is set to continue with new, ever more sensitive infrared-capable telescopes, such as JWST and 20–40 m class ground-based telescopes. As a result, and in the future, we may obtain a comprehensive understanding of weather and climate in giant planet upper atmospheres in the solar system and beyond.

**Author Contributions:** J.O. designed and wrote the majority of this manuscript, including editing of figures. T.S. added substantial contributions to the text and played a significant role in the design. All authors have read and agreed to the published version of the manuscript.

**Funding:** This research was funded by the Japanese Aerospace Exploration Agency (JAXA) under the International Top Young Fellowship program.

**Data Availability Statement:** Not applicable.

**Acknowledgments:** The authors would like to thank Henrik Melin of the University of Leicester for helpful feedback on this manuscript.

**Conflicts of Interest:** The authors declare no conflict of interest.

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
