# Peer review of "What the Upper Atmospheres of Giant Planets Reveal"

_remotesensing, doi:10.3390/rs14246326_

Round 1

Reviewer 1 Report

This is an informative and well-written review of remote sensing results on the upper atmosphere of the giant planets.  It makes a valuable reference for researchers who are working on or interested in related fields.

Author Response

Thank you for your time in reading and reviewing our manuscript. 

Reviewer 2 Report

Report on “What the upper atmospheres of Giant Planets reveal”

This paper reviewed the remote-sensing result of the upper atmospheres of Giant planets in our solar system and beyond. The paper is well organized and the review is comprehensive. I think it can be accepted after minor revision. The bellowing please find some minor suggestions.

1.       In the section “Future and conclusions”, it would be better to list summary points on challenging and unresolved issues of the upper atmospheres of Giant Planets.

2.       The following are some typos in the paper:

Line 78: It seems that this sentence has not been finished. “benefiting from a high signal to noise …”  

Line 121: a function -> functions

Line 140: change the symbol “;” to “:”

Line 172: what Earth’s atmosphere -> what in Earth’s atmosphere

Line 192: UV emissions are coming from -> where UV emissions come from

Line 217: established -> deduced

Line 219: but instead quantify -> by instead quantifying

Line 220: I don’t understand the meaning of the sentence “These reveal electron densities as a function of altitude” here.

Line 253: in clearly -> clearly

Line 254: driven -> raised

Line 256: between it -> in between

Line 258: showed -> showed that

Line 265: it is -> , it is

Line 286: and -> with

Line 249: is being driven -> are being driven

Line 398: a range -> a range of

Line 438: equatorward winds are free to escape -> equatorward winds are efficient to transport energy

Line 553: I have trouble in understanding the sentence “likely the result of the upper atmosphere cooling by approximately 100K”.

Line 554: signal to noise -> signal to noise ratio

Line 572: the difference -> the difference is

Line 575: This sentence is difficult to understand and should be rephrased.

Line 577: the word “lifetime“ is repeated twice.

Author Response

Thank you for your time in reading and reviewing our manuscript.

1. To your first point, we agree that a section regarding the open questions in the field was warranted, as such we have added approximately an additional page highlighting (as bullet points) 4 of the major open questions in the field that were discussed in the manuscript. We have also re-arranged the section and the title by ordering it conclusions–>future. Thanks for this good suggestion.

2. The simple minor comments not shown below were all corrected. I highlight the actions taken on the remaining comments here:

L78: Now reads...
"Most of this review focuses on what we have learned at Jupiter and Saturn; the closest, brightest and largest of the four giant planets which have been visited by multiple spacecraft. Owing to their proximity and physical size we have been able to observe them in great detail, allowing us to study global phenomena such as Jupiter and Saturn's aurora transmitting global-scale waves of heat toward the equator, or unique interactions such an in-fall of material from the rings of Saturn known as `ring rain'."

L172: Now reads...
"To determine the effect of Earth's atmosphere on infrared radiation as a function of wavelength, we must first measure the spectrum of a standard star. This allows us to identify the wavelengths of infrared light that are absorbed by the atmosphere, and to apply corrections to account for it."

L219+220: Ah, these 2 lines were supposed to be the same sentence. It now reads...
"Radio occultations operate under a similar principle, but instead quantify the attenuation of a spacecraft's own radio emissions at radio telescopes back on Earth, which allows for electron densities to be determined as a function of altitude (Kliore+2009)"

L553: Now reads...
"likely because the upper atmosphere cooled by approximately 100 K and the emissions are directly proportional to temperature."

L575: Now reads...
"The equatorial influx driven by collision drag was determined to be much higher at 4,800-45,000 kg s-1, with the range of values encompassing variations in the amount of influx as a function of longitude"

Reviewer 3 Report

This is a very well written compilation of today's knowledge about the four gas giants in terms of their atmospheres.

I would suggest a re-arrangement of the indivudual chapters:

Background -> remote sensing methods -> weather & circulation -> aurorae -> ring rain

I case you might be willing to spent some more time: Include a chapter with comparison between the terrestrial planets upper atmospheres (ionosphere/plasmasphere) with those of the giant planets. It does not make much sense to compare the deeper (neutral) sections, since chemistry and composition as well as physical conditions are way different between gas planets and terrestrial planets, however the ionized upper parts share some similarities.

At the end of the day, atmosphere physics provides most benefit if you include all bodies that we know of (solar system so far, maybe even exoplanets in the future).

I leave this up to you. You can also do this in a separate paper, and I think that will be a good one, too.

Author Response

Thank you for your time in reading and reviewing our manuscript.

We appreciate your suggestion on re-ordering and have made some corrections in light of them:
Chapter 2 ("Giant planet upper atmospheres") defines all the basics about ions, neutrals and electrons, and as such needs to precede the topic of "Remote sensing" because remote sensing talks about specific elements within Chapter 2. In order to make this clear, I changed the chapter headings in order to properly define their themes:
"2. Giant planet upper atmospheres" ->"2. Introduction to giant planet upper atmospheres"
"Giant Planet upper atmospheres"->"3. Observing giant planet upper atmospheres"

We also must still start with "Giant planet aurorae" because otherwise we are describing the circulation of something that the readers do not know much about. In order to give additional context to the readers, we added "Aurorae were the first feature observed in giant planet upper atmospheres owing to their bright emissions." at the start of "Giant planet aurora". This way, the readers will know we are starting with the most obvious feature first and also allowing us to proceed through the paper in order of discovery.

To your second point, I understand what you mean. We understood that the special issue to which we are submitting is chiefly on giant planets, but we did cross-reference auroral physics at Earth and Jupiter (dawn storms in UV at both planets) and also use our knowledge of Earth's upper-atmospheric temperature as our base for describing the energy crisis affecting all giant planets. We hope this is good basic context for now. I think that a terrestrial/giant world comparative planetology style paper would be valuable and interesting in future, helping to bridge the gap between the fields. I also think that such a paper would be better led by experts from multiple planets, rather than myself and Tom who are but humble giant planet experts.

Thanks again for your suggestions.